# Transcriptome Profile Alterations with Carbon Nanotubes, Quantum Dots, and Silver Nanoparticles: A Review

**DOI:** 10.3390/genes12060794

**Published:** 2021-05-23

**Authors:** Cullen Horstmann, Victoria Davenport, Min Zhang, Alyse Peters, Kyoungtae Kim

**Affiliations:** 1Department of Biology, Missouri State University, 901 S National, Springfield, MO 65897, USA; Horstmann95@live.missouristate.edu (C.H.); Davenport17@live.missouristate.edu (V.D.); Zhang1996@live.missouristate.edu (M.Z.); Alyse625@live.missouristate.edu (A.P.); 2Jordan Valley Innovation Center, Missouri State University, 542 N Boonville, Springfield, MO 65806, USA

**Keywords:** next-generation sequencing, engineered nanomaterials, carbon nanotubes, quantum dots, Ag nanoparticles, transcriptomic

## Abstract

Next-generation sequencing (NGS) technology has revolutionized sequence-based research. In recent years, high-throughput sequencing has become the method of choice in studying the toxicity of chemical agents through observing and measuring changes in transcript levels. Engineered nanomaterial (ENM)-toxicity has become a major field of research and has adopted microarray and newer RNA-Seq methods. Recently, nanotechnology has become a promising tool in the diagnosis and treatment of several diseases in humans. However, due to their high stability, they are likely capable of remaining in the body and environment for long periods of time. Their mechanisms of toxicity and long-lasting effects on our health is still poorly understood. This review explores the effects of three ENMs including carbon nanotubes (CNTs), quantum dots (QDs), and Ag nanoparticles (AgNPs) by cross examining publications on transcriptomic changes induced by these nanomaterials.

## 1. Introduction

Sequence based high-throughput technology has made it possible to observe gene expression changes and is quickly becoming the preferred choice in toxicological studies through identifying the modes of action (MOA) or the mechanisms of toxicity of certain chemicals and materials. These technologies are extremely sensitive and capable of whole-transcriptome sequencing and can quantify gene expression alterations by measuring RNA transcripts. The most accurate technique, RNA-Seq, has been around for more than a decade and can detect the slightest changes in gene expression with unprecedented accuracy [1]. It is quickly surpassing older technological sequencing methods including microarray in gene expression studies. RNA-Seq has been shown to provide gene expression measurements with greater quantitative and qualitative information and with greater accuracy than the microarray method [2]. Furthermore, RNA-Seq technology, coupled with bioinformatics tools, has been shown to be the superior approach in studying global gene expression dynamics in practically all biological settings [2]. For this review, we gathered and present findings from previously published studies that investigated the toxicity of engineered nanomaterials (ENMs) using gene expression techniques such as microarray and RNA-Seq.

ENMs include a wide variety of nano-sized particles with incredibly small size ranges of 1–100 nm [3] that possess extremely diverse characteristics. There are many different classes of ENMs including single (SWCNTs) and multi-walled carbon nanotubes (MWCNTs), gold (AuNPs), and silver nanoparticles (AgNPs), fullerenes, dendrimers, metal oxides, quantum dots (QDs), and several more [4,5]. Since their discovery, our understanding of nanotechnologies has broadened, resulting in novel and innovative medical and commercial applications. ENMs have attracted the attention of nearly every industry including the chemical, catalytic, electronic, optical, cosmetic, food, and medical industries [3,6,7,8]. For instance, some of their applications include the diagnosis and treatment of a considerable number of human diseases [3]. Even so, there is still much we do not understand regarding their potential toxicity such as their MOA in biological systems. However, if we want to implement ENMs into products and applications responsibly, more research must be conducted to better understand their unpredictable, and potentially negative, effects.

It has been commonly established in ENM-toxicity studies that their physiochemical properties contribute to their toxicity. The high diversity in physiochemical properties amongst ENMs has made characterizing their modes of action or identifying their individual mechanisms of toxicity in biological systems exceedingly difficult [9,10]. In addition to these challenges, the physiochemical properties of ENMs have been found to change after interacting with biomolecules [5], which add to the complexity of their toxicity so that some ENMs may show greater toxicity in certain biological systems compared to others. Furthermore, there have been drastically different reports on ENM-toxicity among ENMs with nearly identical physiochemical properties and chemical composition in similar biological models [10]. We must overcome such limitations in understanding ENM toxicity if we ever hope to establish regulatory methods for determining the potential risks of ENMs on public health and the environment [11].

In this review, we attempt to examine the toxicity of three ENMs including AgNPs, carbon nanotubes (CNTs), and QDs. AgNPs and CNTs were selected because silver is the most common nanomaterial used in products, followed closely by carbon-based nanomaterials [12]. Additionally, QDs were selected because of their superior applications in biological imaging compared to current dyes [12], and their promising applications in diagnosing and treatment of many diseases. Our aim was to provide and discuss the most up-to-date findings on the toxicity of these materials from studies that utilized toxicogenomic approaches versus studies that implemented toxicological assays, and by highlighting and cross-examining their effects on global gene expression profiles in multiple cellular and higher organism models.

## 2. RNA-Sequencing and DNA Microarray Techniques

DNA microarray and RNA-Seq are the best methods for whole transcriptome gene expression profiling in toxicogenomic studies [13,14]. DNA microarrays utilize nucleic acid probes that are attached to a slide that hybridize with fluorescently labeled DNA target sequences that are then scanned and processed [13]. DNA microarray is a reliable method and more cost effective compared to RNA-Seq. However, RNA-Seq has been found to be more quantitatively accurate than DNA microarray and has become the favored gene expression profiling method. RNA-Seq works by fragmenting RNA prior to converting it into cDNA. Furthermore, the cDNA fragments are labeled with specific adapter sequences and then full transcripts are assembled and counted [13].

The limitations of DNA microarray are apparent when target sequences are composed of unknown genomic sequences. This method only works by using nucleic acid probes of known sequences that are covalently bound to a slide. Therefore, microarray techniques are only as accurate as the bioinformatic data available on an organism’s genome [13]. In comparison, RNA-Seq techniques are also capable of detecting previously annotated transcripts, but can also detect novel sequences, splice variants, non-coding RNA (ncRNA), single nucleotide polymorphisms (SNPs), fusion genes, and using the data from the same experiment can characterize exon junctions [13,14]. Additionally, RNA-Seq can examine transcripts of organisms without a reference genome such as many species of bacteria, and can be used to distinguish between host and parasite transcripts. Furthermore, RNA-Seq can detect significantly lower changes in gene expression compared to DNA microarrays [13,14]. However, RNA-Seq has a few disadvantages compared to DNA microarray such as a lack of standardized protocols for analyzing the data, even though there are plenty of computational tools available. Another disadvantage of RNA-Seq is their considerably larger file sizes compared to DNA microarray and requires more complex bioinformatics analysis, and the computational software can get very expensive [14].

RNA-Seq is clearly the superior technique for gene expression profiling, however, DNA microarray is still valid and implemented in gene expression investigations. Due to the rapid advancements in software, RNA-Seq will eventually become more cost effective. Currently, the use of DNA microarray and RNA-Seq in evaluating ENM-toxicity is not very common, but is becoming more frequent, RNA-Seq more so than DNA microarray. Notably, DNA microarray techniques will not become obsolete anytime soon partly due to their clinical applications in disease diagnostics [14]. Soon, RNA-Seq will likely be used more routinely by researchers as it becomes more cost effective, while DNA microarray will likely be limited to a handful of predetermined uses [13].

Due to the evident advantages of using RNA-Seq in identifying changes in gene expression, toxicogenomic techniques have become a useful tool for assessing risk to human health [15]. These techniques are clearly superior to toxicological assays, which are less revealing in determining mechanisms of toxicity, and provide greater insights on ENM-induced mechanistic changes and their MOA in biological systems [15]. With toxicogenomic data, hundreds or thousands of differentially expressed genes (DEGs) can be identified in organisms exposed to ENMs and new hypotheses can be formed based on the phenotypes those DEGs are implicated in. Therefore, the scope of this review is to collect and organize previously published toxicogenomic results on AgNP, CNT, and QD toxicity in a variety of model organisms. Our goal was to provide the most up-to-date findings on gene expression profiles in ENM-exposed organisms so that future researchers can follow existing leads and continue investigating their toxicity.

## 3. Carbon Nanotubes

CNTs are a highly versatile class of ENMs that have a variety of uses ranging from high performance batteries and touchscreens to drug delivery systems (Appendix A) [5]. They can be described as nanoscopic cylinders that resemble chicken wire and typically stand shoulder to shoulder in a “nanotube forest” that is stronger and lighter than titanium. Their superior tensile strength come from strong bonds between carbon atoms composing the nanotube structure. Due to their excellent physical properties, CNTs were used for many applications such as cancer therapy, bone cell proliferation, bone formation, novel battery technologies, photovoltaics, and nano-based transistors [16]. Their diverse capabilities have led to a fast-growing market for CNTs, however, research on their mechanisms of toxicity are debated and not complete. As a result of increased CNT manufacturing, there is also a strong possibility of increased CNT exposure on the environment. Unfortunately, their nano-size makes them unpredictable and research on their biomechanics and toxicity in humans is limited [17]. Until there is a better understanding of the effects of CNTs on human health, their efficacy in medicine and safe use in manufactured products will remain unknown.

Previous studies have shown that exposure to CNTs can have toxic effects through the generation of oxidative stress or by inducing inflammation [18]. MWCNT and SWCNT are two important classes of CNTs, and their underlying mechanisms of toxicity have been investigated using RNA-Seq. To have a complete understanding of carbon nanotube function in vivo, determining their structure is essential. Nano-carbons are built from sp2 hybridized carbon atoms, a strong molecular interaction that makes CNT more appealing in situations such as drug delivery (Appendix A) [6]. The tubular structure of SWCNT depicted each carbon joined by three neighboring carbons, essentially making a Sp2 hybridized structure [7]. The Sp2 structure of CNTs is more effective than the Sp3 hybridized structure t found in SWCNT. The Sp3 structure can cause deformations by bending or twisting of the nanotube on the wall [7]. The Sp2 structure found in CNTs reflects the structure of graphene, providing unique strength compared to the contorted structure of sp3 hybridized elements. Along with deformations, CNT size has also raised concerns. SWCNTs can have a diameter as small as 0.4 nm and MWCNT can range from 5–100 nm [6]. Smaller CNTs can lead to an increased surface area and a greater potential opportunity for interaction and uptake by living cells [8]. Most studies are in agreement that the smaller size of SWCNTs show a greater cytotoxic effect compared to MWCNTs [19].

CNTs have been shown to have toxic effects on bacterial cells when treated at high concentration. There are four studies, all showing that both SWCNTs and MWCNTs are cytotoxic at higher concentrations in bacteria. To investigate the changes in protein and gene expression in response to SWCNTs, 10 and 100 μg/mL pristine SWCNTs, and hydroxyl and carboxylic functionalized SWCNTs were treated to *Escherichia coli* (*E. coli*) [20]. A direct relationship between cell damage and concentration of SWCNT was observed in *E. coli* cells [9] at low concentrations (10 μg/mL) of SWCNTs, *E. coli* produced phage shock pathway, and altered protein regulations; at the high concentration (100 μg/mL) of SWCNTs, several proteins were shut down [9]. An older study, though still applicable, mirrored the same results. A global gene analysis of RAW246.7 cells treated with 0–50 µg/mL SWCNTs was conducted and the results showed altered gene expression related to the ribosome. Both studies emphasized the toxicity of the CNTs at high concentrations that influence protein production/function. The toxicity of the CNTs was analyzed further, in studies below, influencing different gene pathways.

Zheng et al. investigated the effects of SWCNTs on bacterial growth and had similar results. The result showed that there was an inhibiting effect on *P. denitrificans* bacterial growth rates and densities at the higher concentrations (10 and 50 mg/L) of carboxyl-modified SWCNTs due to downregulated genes associated with glucose metabolism. Carboxyl-modified SWCNTs are transcriptional activators of genes encoding nucleotide reductases. These reductases respond to DNA damage and reduce gene expression and energy production associated with glucose metabolism. Furthermore, carboxyl-modified SWCNT leads to significant downregulation of nitrate reductases, reducing their activity [11]. These results highlight the influence that physiochemical properties have on toxicity. Each CNT (modified and non-modified SWCNTs), though extremely similar in structure and composition, had very different effects on growth.

CNTs have been explored as an antibacterial agent when treated to food pathogen. CNTs can impose a variety of complications on a bacterial cell such as interruption of electron transfer, disruption of cell membrane and cell wall, DNA damage and oxidative stress [10]. Because of these negative side effects, CNTs are investigated as having antibacterial effects. Gene expression studies on bacteria, specifically *Salmonella typhimurium*, can help clarify the mechanisms of CNT toxicity and if this treatment could be a possible anti-bacterial agent. *Salmonella typhimurium*, a Gram-negative food borne pathogen, was exposed to SWCNTs in order to study the gene associations with bacterial metabolism, structural integrity, and antibacterial components of nanoparticles, using electron microscopy and molecular studies such as qRT-PCR. Two silver coated carbon nanotubes, SWCNT-Ag and pSWCNT-Ag (pegylated SWCNT-Ag) are predicted to have antibacterial activity mediated through the generation of ROS from the bacterial cells [12]. The genes associated with the bacterial membrane were upregulated when treated with SWCNT-Ag but downregulated in response to pSWCNT-Ag. In summary, pSWCNT-Ag are non-toxic to human cells compared to SWCNTS-Ag. Experiments show that pSWCNT-Ag has been proven to be a potential safe alternative antimicrobial agent to treat food borne pathogens.

In 2020, Elsehly et al. found that higher concentrations (80 and 60 μg/mL) of functionalized multiwall carbon nanotubes (F-MWCNTs) showed maximum bacterial inhibitions and antibacterial functionality on *E. coli* and *S. aureus* due to the cell death from the oxidative stress. The antibacterial activity of F-MWCNTs was implicated in biomedical devices and cleaning applications of hospital and industries [21]. Antibacterial effects and toxicity of CNTs have led researchers to experiment on cancer cell lines.

CNTs have been researched in pathways related to cancer, especially lung cancer. In 2012, Guo et al. investigated MWCNT-induced gene signature in mouse lungs and the association between these genes and human lung cancer risk and prognosis (Appendix A). Mice were treated with MWCNTs at concentrations of 10, 20, 40, and 80 µg and RNA was extracted for microarray. Twenty-four genes were selected that had a significant change in at least two time points. At day 56 after exposure to MWCNTs, 330 genes were differentially expressed, and 38 of them were related to cancer [13]. CNTs can cause cancer mutations. A closely related study conducted with mouse lung revealed similar effects: a subset of mouse lung cancer biomarkers was affected after exposure to MWCNTs. Pancurari et al. found that seven of a total 63 lung cancer prognostic and major signaling biomarker genes had different expression levels compared to the control group at seven days after exposure to MWCNTs, and 11 genes had different expression levels compared to the control group at 56 days after exposure to MWCNTs by using qRT-PCR (Appendix A). Additionally, among the 11-gene associated canonical pathways, the molecular mechanisms of the cancer pathway ranked the most significant at 56 days after exposure [14]. Guo and Pancurari both emphasized the potential harmful effects of MWCNTs when treated to mouse model organisms. These studies convey the influence of MWCNTs on cancer related pathways when gene expression studies were conducted. MWCNTs have obvious adverse effects, but SWCNTs could have different results because of the smaller size.

SWCNTs have one wall surrounding their structure, making them potentially more toxic and easier to penetrate the cell compared to MWCNT. Inhalation of these products is a possible concern, so studies are conducted with SWCNTs to show effects. Fujita et al. investigated time dependent changes in gene expression associated with the pulmonary toxicity of SWCNTs. SWCNT suspensions were administered one time in each rat (0.2 mg or 0.4 mg). At 90 days after SWCNT exposure, the persistence of macrophages laden with SWCNT aggregates were observed in the alveolar walls and alveoli [15]. At 180 days, they observed macrophage-containing granuloma around the sites of SWCNT aggregates. Additionally, many genes involved in inflammatory response were significantly upregulated on days 7, 90, and 180 and the number of upregulated genes gradually decreased 180 days after instillation, but increased again at 365 days [15]. Overall, SWCNTs have been shown to aggregate in the alveolar walls initiating pulmonary effects. Genomic studies experimenting on different human epithelial cells are most relevant when investigating biomarkers for human lung disease and other side effects.

Although CNTs are the popular candidates for creating new and better products, their adverse effects on humans through inhalation and dermal contact is still a concern. A previous study in 2014 investigated the correlation and concordance of MWCNT-induced gene expression in vitro monoculture and co-culture of human small airway epithelial cells (SAEC) and human microvascular endothelial cells (HMVEC) with gene expression in vivo mouse lung exposed to MWCNT. Snyder-Talkington et al. compared MWCNT-induced mRNA gene expression from human SAEC and HMVEC in monoculture and coculture [18,22]. When HUMVEC were cocultured and treated with MWCNTs, there were more concordant genes (both up- or downregulated in vivo and in vitro) than those of monoculture, particularly disease-related concordant genes [19,22]. Since the gene expression of the coculture model correlated better to the in vivo gene expression, MWCNTs could also be the potential biomarker for human lung diseases (Appendix A). A similar study was conducted in 2019, where Snyder-Talkington et al. found concordant mRNAs—LC7A1 and SLC22A5 were downregulated in all mice and human tissue, blood, and cell analyses, which could cause human primary hypertension, cardiovascular diseases, encephalopathy, cardiomyopathy, cardiomegaly, metabolic derangement, hypoglycemia, and muscle weakness (Appendix A) [18].

The treatment of CNTs on human cells causes many different complications: hypertension, cardiovascular disease, and much more. Increased ER stress is a major factor that relates to these issues. Treatment of CNTs on human cells was conducted in a 2017 study. RT-PCR was used to quantify the mRNA level of *ddit3(chop)* and *xbp-1s*, two gene biomarkers for ER stress. A recent study found that ER stress caused possible dysfunction of endothelial cells via the ER stress pathway when exposed to CNTs (Appendix A) [16]. Human umbilical vein endothelial cells (HUVECs) were treated with 32 mg/mL of XFM22 (shorter MWCNT) and XFM19 (longer MWCNT) [16]. Using specific primers for each gene, mRNA was quantified. Exposure of XFM22 to the HUVECs decreased the expression of *ddit3* in mRNA whereas when HUVECs were treated with XFM19, the cells expressed a significant increase in *ddit3*. Longer MWCNTs showed more ER stress then shorter MWCNTs [16]*. Ddit3* is a transcription factor that regulates inflammatory cytokines like IL-6. XFM19, the longer MWCNT, increased *ddit3* expression, meaning that XFM19 treated HUVEC cells could release IL-6, therefore increasing the expression of *ddit3.* Increasing the expression of ddit3 resulted in upregulated inflammatory response when treated with the XFM19 longer MWCNT. XFM19 elevated both ER stress and inflammatory response compared to the XFM22 shorter variant. The influence of MWCNT on gene expression associated with ER stress in HUVECs was also investigated in 2018. Chang et al. found that the expression of genes associated with ER was induced by high concentrations of MWCNTs in HUVECs cultured in the upper chambers such as HSPA5, DDIT3, and XBP-1s (Appendix A) [19]. According to the results, they all provided evidence of CNTs negatively affecting HUVECs via ER damage. Interestingly, Sager et al. found that carboxylate (COOH) groups of functionalized MWCNTs significantly reduced the lung inflammatory responses after pharyngeal aspiration, likely due to decreased association with lung cells [23].

In 2010, Patlolla et al. tested the potential effects of MWCNTs on normal human dermal fibroblast (NHDF) cells based on three doses: 40, 200, and 400 μg/mL [18]. After exposure to different concentrations of MWCNTs, cell viability showed a significant dose-dependent reduction [18]. Additionally, they tested genotoxicity and apoptosis for NHDF cells and found a direct correlation between the concentration of MWCNTs and percentage of apoptosis as well as the levels of tail DNA, which is a parameter in evaluating cell DNA damage [17]. In another study, Siegrist et al. also tested the genotoxicity effects of MWCNTs on the cultured primary and immortalized human airway epithelial cells, and their results showed that there was an increase in spindle disruption, abnormal mitotic spindles, and aneuploid chromosome number with the increased doses of MWCNTs [18]. Furthermore, in 2019, Snyder et al. employed qPCR to measure the potential effects of MWCNTs on mitochondrial gene expression in human bronchial epithelial cells (BECs) after exposure to up to 3 μg/mL of MWCNTs. As a result, the mitochondrial gene expression in some BECs was significantly upregulated [6,19]. All data demonstrated that the levels of DNA and mitochondrial damage in human cell lines were increased by MWCNTs. The exposure of MWCNTs to humans could cause adverse health effects in a dose-dependent manner.

In conclusion, the potential effects of CNTs on gene expressions was studied to promote the application of nanotechnology and the development of nanoparticles. Although they were a very popular nanoparticle for biomedical applications, several studies have also demonstrated that CNTs can be harmful by affecting gene expression and protein pathways. CNTs led to upregulation of gene expression for oxidative stress, inflammation, and cancers [21,24,25,26]. These articles provided empirical evidence and support for an accurate assessment of the potential risks in bacteria, animal, and human models that will help us to better understand their long-term effects on our health and the environment. The effect of CNTs on gene expression may serve as a potential biomarker for human medical and occupational monitoring in future studies. Due to the lack of research on gene expression, the production and use of CNTs in biomedical technologies such as disease diagnosis and drug delivery are still at risk. Hopefully, we can use them responsibly after investigating their unique properties on gene expressions further.

## 4. Quantum Dots

QDs are a newer class of ENMs with a diameter of 2–10 nm [27,28] that possess unique physical and chemical properties that include high stability, narrow emission ranges, and high quantum yield [29]. Therefore, QDs are broadly used in biosensors, molecular imaging, multicolored labeling, drug carrier cosmetics, therapeutic targeting, photodynamic therapy, and real-time tracking [29,30]. Most QDs are considered non-toxic and as a result, are of enormous interest in chemical, catalytic, electronic, optical, mechanical, magnetic, and medical fields and are becoming more common in many diverse commercial and industrial sectors including, but not limited to, textiles, medical products, cosmetics, paints, and plastics [3]. Similar to CNTs, QDs are a potential smart drug delivery vehicle in new and innovative cancer treatments [31]. Their photo-stability, tunable emission, and broad excitation range make them a more effective fluorescent tag than organic dyes in biological applications (protein labels, real-time trackers, and FRET sensors) [24,25]. Previously, there have been conflicting results regarding QD cytotoxicity due to their diverse physiochemical properties (size, charge, composition, concentration, outer coating bioactivity, and stability), believed to be determining factors of toxicity [9,29,32]. Thus, the high possible combinations of properties result in a wide possibility of cytotoxic effects [24]. Cytotoxicity has been investigated extensively; the scope of this section is to provide the most relevant and current findings on the transcriptomic effects of QDs on the cellular and organismal level.

Given the exceedingly broad applications of QDs, it is vital to resolve the uncertainty of their toxic effects in prokaryotic and eukaryotic organisms. To date, very few global transcriptomic analyses of the effects of QD exposure on bacterial toxicity have been reported. A study on the response of *E. coli* to CdTe-GSH QDs (Appendix A) revealed that the same QD displayed differential toxicity based on size. Red QDs were found to be more toxic than green QDs when treated to *E. coli* at MICs of 125 and 2000 μg/mL, respectively. To determine the bacteria’s global response to QDs of both sizes, microarray analysis was used to measure changes in gene expression. A total of 95 genes were altered in response to red QDs while only 42 genes were changed in response to green QDs. Moreover, there were seven genes differentially regulated by both QDs that Gene Ontology (GO) analysis reported were implicated in processes related to transport, biosynthesis, and metabolism. Furthermore, red QDs upregulated genes related to glycolysis and the TCA cycle slightly and transport by nearly 4-fold compared to green QD exposure [30]. Additionally, a transcriptomic response of *Pseudomonas stutzeri* exposed to cationic polythylenimine (PEI) coated CdSe/CdZnS QDs was observed. Changes in seven genes, mostly implicated in denitrification (narG, napB, nirH, and norB) and the upregulation of superoxide dismutase (sodB), suggests the production of ROS as a response to QD exposure [26]. Correspondingly, Yang et al. found *P. aeruginosa* PAO1 exposed to CdSe QDs altered the expression of genes implicated in response to heavy metals and oxidative stress [33].

With RNA-Seq, Horstmann et al. investigated the effects of CdSe/ZnS QDs on the yeast *Saccharomyces cerevisiae* (*S. cerevisiae*) and found exposure to the QDs had resulted in thousands of DEGs most notably involved in rRNA transcription, ribosomal subunit assembly, ribosomal subunit transport, tRNA maturation, and translation machinery assembly [32]. Hosiner et al. used microarray to investigate the effects of several metal ions (CdCl_2_) on *S. cerevisiae* and found antioxidant genes and redox homeostasis genes to be upregulated such as *GRX2* and *TRR1*, *TRR2*, and *TRX3*, respectively [34]. Interestingly, in 2016, researchers exploited a mutant *S. cerevisiae* strain to identify the genetic basis of CdS QD resistance. They found that in response to CdS QD exposure, several metabolic processes were altered including abiotic stress response, mitochondrial organization, transport, and DNA repair [35]. The same research team later conducted a transcriptomic analysis and found mitochondrion organization as the primary functional category affected genes fell under. Additionally, they observed diminished oxygen consumption, cytochrome content, and mitochondrial membrane potential [36].

A recent study on soybean tissue investigated the effects of CdS QDs on transport proteins and biological pathways. Majumdar et al. (2019) identified 1690 proteins that were common between each CdS-QD treatment and GO-term analysis suggests the affected proteins in CdS-exposed soybean roots are localized in the cell wall, extracellular region, membrane-bound organelles, and function as integral components of the membrane [37]. CdS-QD-treated soybean root tissue has been found to significantly alter protein levels involved in transmembrane transport of metal ions or protons, chitin binding, carbohydrate metabolism, and responding to oxidative stress [37]. Interestingly, short-term CdS-QD exposure (14 days) on soybean roots were found to upregulate cytosolic proteins involved in metabolic pathways including glycolysis, the TCA cycle, fatty acid β-oxidation, amino acid biosynthesis, and secondary metabolite biosynthesis. Upregulated cytosolic proteins responsible for converting proteins into useable substrates involved in the TCA cycle were unique to CdS-QD exposure. Additionally, downregulated proteins involved in glycogen metabolism (such as uridine-triphosphate- (UTP) glucose-1-phosphate uridylyltransferase involved in uridine diphosphate-glucose (UDP-glucose) regeneration from glucose-1-phosphate) were identified, suggesting that CdS-QD exposure alters soybean metabolic pathways to favor glycolysis [37]. Similarly, in 2013, Simon et al. utilized RNA-Seq to investigate the effects of CdTe/CdS QDs on green algae and identified, through GO-term analysis, DEGs involved in oxidative stress, redox potential, protein folding, and chaperone activity processes [38].

A 2010 study investigated the geno-toxic effects of green CdSe (455 nm), blue Cd1-xZnxS/ZnS (550 nm), and red CdSe/ZnS (625 nm) QDs on human embryonic kidney fibroblast cells (HEK293) by whole-genome microarray. HEK293 cells were treated with 200 nM, 60 nM, and 10 nM of red, blue, and green QDs, respectively. Interestingly, more genes were upregulated in green and blue QD treated cells and red QD treated samples downregulated more genes. GO-term analysis of DEGs showed that response to wounding, cell stress, apoptosis, and defense response functions were enriched in all three of the QDs tested. The expression of metallothionein superfamily genes were induced when treated with red and green QDs, however, these genes were not affected in blue QD treated samples. Metallothioneins (MTX2a, MTX1h, MTX1g, MTX1f) are metal binding proteins that play a role in detoxification and protect against ROS and are also upregulated when exposed to Cd^2+^ ions [39].

Zhang et al. investigated PEG-silane-CdSe/ZnS QDs on human skin fibroblasts (HSF-42) with a genome-wide expression array analysis at concentrations of 8 and 80 nM. Approximately 50 genes had significantly altered expression levels greater than 2-fold and were found to be involved in carbohydrate binding, intracellular vesicle formation, and cellular response to stress. Interestingly, PEG-silane-QDs were found to downregulate genes involved in modulating the M-phase progression of mitosis, spindle formation, and cytokinesis. However, PEG-silane-QDs did not induce immune and inflammatory responses or heavy metal related toxicity, unlike exposure to CNTs. This study provided evidence that if CdSe/ZnS QDs are appropriately coated, they will have very little impact on HSF-42 cells and PEG-coated QDs do not pose a major threat and reduce the toxicity of CdSe/ZnS QDs [40].

Unlike previous gene expression studies that have mainly focused on fibroblast cell lines, a new 2020 study investigated the effects of Cd QDs on the growth of human cervical cancer cells (HeLa). Hens et al. implemented the RNA-seq method to analyze transcriptomic changes in HeLa cells when exposed to QDs. They identified many significantly up- and downregulated genes and conveniently grouped them based on their functions using GO-terms. When exposed to QDs, they observed upregulated cellular functions in HeLa cells such as anti-apoptotic, anti-proliferative, and anti-tumorigenic functions (Appendix A). Additionally, they identified downregulated functions including pro-proliferation, mitochondrial respiratory chain, detoxification, and receptor-mediated endocytosis. Based on new insights from their transcriptomic analysis, they provide evidence that CdSe/ZnS QDs could be effective as an alternative anticancer drug [41]. A similar study conducted by Davenport et al. [used InP/ZnS QDs on HeLa cells. Their RNA-Seq and gene expression analysis revealed many genes involved in developmental processes including differentiation, tissue and nervous system development, and morphogenesis to be upregulated [42]. They also found major processes such as metabolic and biosynthetic processes to be significantly downregulated. Both up- and downregulated processes suggested the expression of pro-apoptotic gene processes and control over cell motility (Appendix A). They, like Hens et al. suggested the QD treatment be considered for anticancer drug development (Appendix A) [42]. Additionally, carboxylated CdSe/ZnS QDs upregulated genes implicated in DNA repair, responding to stress, ATP functions, and RNA activities, and downregulates genes involved in cellular division in alveolar epithelial cells [43].

In summation, Cd based QDs, despite their different spectral characteristics and composition, display toxic effects. Dua et al. and coworkers found with microarray analysis that QD exposure induces the expression of genes involved in oxidative stress, apoptosis, and inflammation, which result in decreased cellular viability. Furthermore, they stated that QD-mediated toxicity is highly dependent on the core material (CdSe) and the coating (ZnS) partially reduces the toxic effects [39]. Their results suggest that the use of unmodified QDs in biomedical applications should be used carefully and with much consideration. In contrast, Zhang et al. showed that by adding a polymer coat such as a PEG-silica-coat, it allowed for their safe use in vivo. They provided evidence with a comprehensive analysis of genome-wide expression alterations that PEG-silica-coated CdSe/ZnS QDs had minimal impacts on cellular health. These results contradict many popular beliefs that CdSe-based QDs are toxic due to Cd^2+^ leakage. These potentially safer polymer coated QDs could be a large step toward their safe and widespread use in biomedical studies and applications [40]. These initial findings provide a solid foundation on QD toxicity necessary in identifying patterns in altered gene expression levels that will ultimately help build a network of global transcriptomic data on lower and higher organisms exposed to QDs. Future studies should focus on QDs’ long-term fate in living organisms including their breakdown to help better fully understand their toxicity. Understanding their effects in biological systems is crucial if we want to responsibly continue to use or incorporate them in industrial or commercial settings in the future.

## 5. Ag Nanoparticles

Silver has been studied and used for thousands of years in the medical and engineering fields because of its antimicrobial properties against bacteria, viruses, and fungi [44,45]. In recent decades, the use of silver has been studied on a micro level in the form of nanoparticles. AgNPs range from 1–100 nm in size and have gained popularity for their optical, thermal, and electrical properties [46,47]. With increasing levels of antibiotic resistance, AgNPs have been hypothesized as a solution for improving antibiofilm hydrogels [48] and current antibiotic treatment options for diseases like tuberculosis (Appendix A) [49]. More specifically, AgNPs have been used to innovate water disinfection, medical diagnostics, pharmaceutical development, anti-cancer therapies, agriculture/livestock treatment, and biofouling control (Appendix A) [50,51,52,53,54].

While silver is found in many daily activities, AgNP production and application have potentially toxic effects to both the environment [55] and the human population. Recently, “green” synthesis of AgNP using plants and microorganisms has been explored as an eco-friendly alternative (Appendix A) [55,56,57,58,59]. In humans, AgNP exposure has been linked to disrupting the function of mitochondria, sperm cells, and cytokine expression [47]. There are also reports of cell defects and induced inflammation [60,61]. Studies on AgNP toxicity are widespread in focus; we were specifically interested in the effect of AgNPs on gene function and transcriptome alterations across species. The scope of this section will summarize the current findings on the effect of AgNPs on the transcriptome of different organisms.

As mentioned, AgNPs are known for having antimicrobial/bacterial activity and have been widely studied against various bacterial strains. It has been hypothesized that AgNP could decrease antibacterial activity to be an alternative to antibiotics in treating bacterial infections [62]. To test this, Ashmore et al. studied AgNPs against *E. coli* using qRT-PCR. Supporting the proposed hypothesis, their findings show AgNPs to significantly downregulate genes associated with the TCA cycle (aceF, gadB) and amino acid metabolism (argC, metL, gadB), pointing to effective antibacterial properties of AgNPs (Appendix A). The downregulated genes positively correlated with the proposed hypothesis, however, there was also an upregulation in genes relevant to bacterial virulence (fliC, msbB) and DNA repair mechanisms (mfD) [62]. Similarly, antimicrobial effects of AgNP have been studied against bacteria (*Staphylococcus aureus* and *Staphylococcus epidermidis)* that produce biofilms and cause biomaterial-related infections in surgically inserted devices [63]. Through qRT-PCR methods, genes implicated in biofilm formation (icaA and icaR in *S. epidermidis*, fnbA and fnbB in *S. aureus*) were significantly downregulated with the presence of AgNP; these results conclude that AgNPs inhibit the transcription of biofilm related genes and to be inhibitory toward *S. aureus* and *S. epidermidis* bacteria (Appendix A) [63]. An important regulator of *S. aureus* growth is found in small regulatory RNAs (sRNA) that require Hfq protein to mediate sRNA and their target mRNA [64]. Targeting the activity of Hfq with AgNPs for antibacterial treatment was studied by Tian et al. in which they specifically investigated sRNA-TEG49 expression (a key mediator of Hfq) [65]. Once again, qRT-PCR was used along with high-throughput RNA sequencing and northern blot analysis. AgNP exposure to *S. aureus* resulted in the loss-of-function of Hfq and subsequent inhibition of sRNA-TEG49. These results suggest that the regulation of Hfq function with AgNPs are important to the NPs’ antibacterial mechanisms (Appendix A) [65]. Collectively, applications of AgNPs against different bacteria support claims that they are effective in altering gene function to control bacterial growth.

Many different fungal species are involved in pathogenesis affecting mammalian life and need effective prevention and treatment methods. Along with their antibacterial properties, AgNPs have also been investigated for potential antifungal properties. Some fungi have been investigated for their production of aflatoxins (AF) such as *Aspergillus flavus* and *Aspergillus parasiticus*. Aflatoxin B_1_ (AFB_1_) has been identified in these fungi and classified as a group 1 carcinogen to mammals; the use of AgNPs as treatment to inhibit AFB_1_ production is proposed by Deabes et al. [66]. It was previously reported that there are three main genes responsible for aflatoxin biosynthesis: *aksA*, *ver-1*, and *omt-A* [67]. Within their study, Deabes et al. measured the expression levels of these three genes, along with the *AFB_1_* gene itself, in *A. flavus* ATCC28542 using qRT-PCR. Their findings showed inhibited *AFB_1_* and *omt-A* expression, leading them to conclude that AgNPs are effective in preventing AF production by *A. flavus* [66].

In some fungi, synthesis of structural molecules aid in their pathogenicity, as seen in *Bipolaris sorokiniana* and melanin production [68]. Specifically related to melanin production, expression of genes *viz*, *PKS1*, and *SCD1* were observed using qRT-PCR. Downregulation of genes *PKS1* and *SCD1* were observed in *B. sorokiniana* exposed to AgNPs, concluding the treatment to reduce melanin synthesis (Appendix A) [68]. These results suggest AgNPs are viable in antifungal treatment, but the authors note that further investigation is needed to understand the correlation between pathogenicity and melanin production in *B. sorokiniana*.

In addition to pathogenic fungi, other related species have commonly been used as model organisms for studying biological processes. *Szhizosaccharomyces pombe* (fission yeast) is commonly used to study cell morphogenesis and division; after constructing a genome-wide deletion library of fission yeast, Lee et al. used the organism to identify target genes for tolerance against AgNP-induced cytotoxicity [69]. Target screening and q-PCR were used to identify seven nonessential genes related to sulfur metabolism (*gcs1, gcs2, hmt2,* and *rdl2*) and MAPK kinase signaling (*mcs4, wis4* and SPCC1827.07*c*), all of which were previously linked to metal resistance and stress response. Three essential genes (*met9, sfh1,* and *peg1*) related to carbon metabolism were linked to AgNP-induced cytotoxicity for the first time (Appendix A) [69]. The findings of this study are important to understanding genetic defenses against AgNP cytotoxicity within fission yeast.

Other fungal species such as *Folsomia candida* and *S. cerevisiae* is often studied because they culture easily and grow quickly. Both *F. candida* and *S. cerevisiae* were studied by Sillapawattana et al. to understand the molecular toxicity of AgNPs and propose their relevance as eukaryotic model organisms in ecotoxicological testing. Target genes *GST* and *MT* were measured using qRT-PCR; *F. candida* observed upregulation of target genes when exposed to AgNPs, demonstrating the organism’s ability to respond to a changing environment and use employed gene expression to examine the chemical effects in toxico-genomic studies (Appendix A) [70]. Observing the yeast genome showed code for only one cytosolic and one mitochondrial enzyme. This exemplifies the handling of yeast for chemo-genetic screening to understand AgNP toxicity in yeast [70].

In our own lab, AgNP toxicity in *S. cerevisiae* was also investigated by Horstmann et al. After observing decreased viability following AgNP treatment, RNA-seq was used to identify genetic alterations to the yeast (Appendix A). Specifically, upregulated gene processes are suggested to disrupt healthy ribosome function and successive rRNA/tRNA synthesis [71]. The study also analyzes downregulated processes that point to a defect in cell wall organization [71]. Conclusively, AgNP toxicity in yeast appeared to be heavily influenced by gene expression alterations.

Expanding from bacterial and fungal treatments, AgNPs have also been proposed for medical therapies and technologies. To understand the positive and negative effects of using AgNPs in humans, scientists have turned to various human cell lines and mice models for answers. Within these efforts, it is important to focus on gene expression changes to determine how the NPs interact with the cell. Using RNA-seq, Gurenathan et al. observed changes to gene expression of in vitro NIH3T3 mouse embryonic fibroblasts. Alteration of processes involving epigenetics such as nucleosome assembly and DNA methylation were found when treated with AgNPs. The study also found increased levels of apoptosis, which the authors suggest is influenced by a repression of genes related to cell survival (Appendix A) [72]. In a similar model, mice neural cells were observed with AgNPs to relate gene expression changes to the development of neurological disorders like Alzheimer’s disease. RT-PCR and western blot revealed increased gene expression related to the amyloid β peptide responsible for causing Alzheimer’s (Genes *GSS*, *CYCL13*, and *MARCO*) (Appendix A) [73]. As well as inducing immune reactions and stress-response within the neural cells, the conclusions from this RT-PCR analysis suggests that AgNP exposure accelerates the formation of plaque associated with Alzheimer’s disease and emphasizes a need to monitor our daily interaction with Ag.

The toxicity of AgNPs is prevalent, however, its properties also have the potential to be productive. The use of AgNPs as an anti-cancer treatment shows the potential to apply toxic properties to cancer cells to destroy them. This was observed in combination with another cancer treatment, campotothecin (CPT), in cervical cancer cells (HeLa) by Yuan et al. The combined treatments increased the expression of pro-apoptotic genes like *p53*, *p21*, *Cyt C*, *Bid*, *Bax*, and *Bak*; modification of signaling molecule expression related to cell survival, viability, and cytotoxicity were also observed (Appendix A) [74]. NGS has also been used to show the negative effect of AgNP treatment on human lung cells; evidence of DNA damage has raised concern for human exposure in vitro [75]. It is suggested that the combination of these therapies, at low enough doses, could be successful at inducing apoptosis in cancer cells without causing unwanted cytotoxic effects [74].

The study of AgNPs in human tissue is crucial to discovering the strengths/limitations of their application to human cancer. Opposed to their anti-cancer potential, recent evidence has shown that AgNPs alter genetic expression to increase susceptibility to carcinogens (Appendix A) [76]. Our understanding of the long-term in vitro effects on carcinogenicity can be improved by future studies of these nanoparticles [75]. The future use of genomic studies will be valuable to advance the applications of AgNPs in the oncology field.

In conclusion, the use of AgNPs has greatly impacted antibacterial, antifungal, and anticancer treatment methods. This review only touches the surface of data that has been reported, but shows evidence concluding that AgNPs are effective in the alteration of gene expression among species. While its properties are promising, careful consideration is still needed in reference to their toxicity within mammalian cells. Going forward, more research is needed to understand what levels of AgNPs are safe for human consumption and ways to harness their toxicity so that they can continue to be an effective treatment against pathogenesis.

## 6. Conclusions

Toxicogenomic techniques such as DNA microarray and RNA-seq could be valuable tools for assessing toxicity. However, RNA-seq provides more precise coverage of the transcriptome of a cell [14]. Due to its wider dynamic range and it superior coverage of DEGs, RNA-seq could be a great tool for understanding toxic materials and their mechanisms of toxicity. However, these modern genomic approaches are not without their limitations. There are still not many standardized protocols on the bioinformatics analysis required for RNA-seq [14]. Additionally, computational software can range from free open-source online platforms to software with drastically different capabilities that cost several thousands of dollars. Despite the technical challenges, DNA microarray and RNA-seq are still both suitable for toxicogenomic investigations that give us much insight into the transcriptomes of ENM-exposed cells and their risk to public health and the environment.

## Data Availability

Not applicable.

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
