# Peer review of "Transcriptome Profile Alterations with Carbon Nanotubes, Quantum Dots, and Silver Nanoparticles: A Review"

_genes, 2021, doi:10.3390/genes12060794_

Round 1

Reviewer 1 Report

The manuscript is outstanding. In literature there is plenty of confusion where the usage of inorganic materials as bio inspired systems. This manuscript will definetly clear out such confusion. It is timely and covers all important topics of sequencing technology.

Author Response

Revisions can be seen on the updated manuscript with track changes. Detailed responses are seen below in blue italicized font. We hope each suggested change is satisfied by our edits and thank you again for the efforts given to improve our manuscript.

Reviewer 1:

The manuscript is outstanding. In literature there is plenty of confusion where the usage of inorganic materials as bio inspired systems. This manuscript will definetly clear out such confusion. It is timely and covers all important topics of sequencing technology.

The authors thank Reviewer 1 for their kind comments and for taking the time to read and edit our manuscript.

Again, we extend our gratitude for the opportunity to improve our manuscript. We hope our efforts reflect the valuable comments from each reviewer. Thus, we would like to resubmit the updated article for further consideration.

Sincerely,

Kyoungtae Kim

Missouri State University

901 S National Ave.

Springfield, MO, USA 65802

Kkim@missouristate.edu

Reviewer 2 Report

In the current form this review does not bring much to the field. It's mainly an enumeration of studies that are poorly described. There are only few references in the introduction and I'm missing important references such as Costa PM, Fadeel B. Emerging systems biology approaches in nanotoxicology: Towards a mechanism-based understanding of nanomaterial hazard and risk. Toxicol Appl Pharmacol. 2016 May 15;299:101-11. doi: 10.1016/j.taap.2015.12.014. Epub 2015 Dec 22. PMID: 26721310.

I'm missing a justification on the selection of the three nanomaterials. Why is it important to have a review on CNTs, Ag, and QD? 

I'm missing a discussion on what these toxicogenomic studies mean? Have they been validated in functional assays, is there a change in phenotype etc? Is there a consensus between these studies? Are these studies capable of predicting adverse outcomes? Have these studies been integrated in the development of AOPs (adverse outcome pathways)? 

I'm also missing a technical discussion on how the data in these studies is analysed, and how can the type of down-stream analyses influence the outcome. Is there a consensus on what is the best way to deal with these big data?

What is the advantage of toxicogenomic approaches compared with traditional toxicological assays? How can these novel approaches can be used for hypothesis generation? 

In Table 1 the findings are very vaguely described and it's difficult for the reader to get an idea on the results. Relevant info should relate to the number of differentially expressed genes, top enriched pathways/networks/gene ontologies depending on the type of analyses. If in vivo studies which tissues have been analysed. Also, have the findings been validated, for example functional validation, protein expression. There should be more info on the articles such as author and journal as well as the methods used and types of down-stream analyses (gene ontology, pathway enrichment analysis). 

The conclusion should focus on how these studies can be valuable for predicting toxicity as well as how these studies can be validated. Sure, proteomics can be of use but is by far the only way to validate this data. It is very important if the changes in gene expression and protein expression can lead to a change in phenotype. 

Author Response

Revisions can be seen on the updated manuscript with track changes. Detailed responses are seen below in blue italicized font. We hope each suggested change is satisfied by our edits and thank you again for the efforts given to improve our manuscript.

Reviewer 2:

In the current form this review does not bring much to the field. It's mainly an enumeration of studies that are poorly described.

We thank you greatly for your time and effort in editing our manuscript. Your insight has been applied to the updated manuscript and we feel the writing has been improved from your suggestions.

There are only few references in the introduction and I'm missing important references such as 

Costa PM, Fadeel B. Emerging systems biology approaches in nanotoxicology: Towards a mechanism-based understanding of nanomaterial hazard and risk. Toxicol Appl Pharmacol. 2016 May 15;299:101-11. doi: 10.1016/j.taap.2015.12.014. Epub 2015 Dec 22. PMID: 26721310.

According to reviewer two’s comment on missing important references in the introduction, the authors have added several more references to the introduction including “Costa et al., 2016”, mentioned specifically. 

I'm missing a justification on the selection of the three nanomaterials. Why is it important to have a review on CNTs, Ag, and QD? 

In accordance with reviewer two’s comment, the authors have added a justification on the selection of the three nanomaterials to the introduction. QDs were selected because of their promising diagnostic applications in biological imaging, and AgNPs and CNTs were selected because Ag is the most common nanomaterial used in products followed closely by carbon-based nanomaterials.

I'm missing a discussion on what these toxicogenomic studies mean? Have they been validated in functional assays, is there a change in phenotype etc? Is there a consensus between these studies? Are these studies capable of predicting adverse outcomes? Have these studies been integrated in the development of AOPs (adverse outcome pathways)?  I'm also missing a technical discussion on how the data in these studies is analysed, and how can the type of down-stream analyses influence the outcome. Is there a consensus on what is the best way to deal with these big data?

The authors have addressed reviewer two’s comment about missing a technical discussion on how the down-stream analysis of these studies (Micro array and RNA-seq) influence the results and if there is a consensus on how to process these big datasets by adding a section after the introduction that addresses how Microarray and RNA-seq meet the demand for reliably assessing transcript abundance in biological samples and using that information to better understand the relationship between ENM-induced transcriptome alterations and the relationship between genotype and phenotype changes in regards to ENM toxicity.   

What is the advantage of toxicogenomic approaches compared with traditional toxicological assays? How can these novel approaches can be used for hypothesis generation? 

Reviewer two’s questions on the advantages of toxicogenomic approaches compared with traditional toxicological assays and how these novel approaches can be used for generating hypothesis was addressed by the authors in the new section (after the introduction) on Microarray and RNA-seq methods. 

In Table 1 the findings are very vaguely described and it's difficult for the reader to get an idea on the results. Relevant info should relate to the number of differentially expressed genes, top enriched pathways/networks/gene ontologies depending on the type of analyses. If in vivo studies which tissues have been analaysed. 

In response to Reviewer 2’s comment, the authors have decided to replace Table 1 with 3 smaller, more descriptive tables. Specific differentially expressed genes are now noted and we believe the tables to be much more informative in their updated state. Thank you.

The conclusion should focus on how these studies can be valuable for predicting toxicity as well as how these studies can be validated. Sure, proteomics can be of use but is by far the only way to validate this data. It is very important if the changes in gene expression and protein expression can lead to a change in phenotype. 

In response to Reviewer 2’s comments on the conclusion, the authors have revised it so that it focuses on the value of DNA microarray and RNA-seq for investigating toxicity. We compare the techniques and discuss their advantages and limitations in the field.

Again, we extend our gratitude for the opportunity to improve our manuscript. We hope our efforts reflect the valuable comments from each reviewer. Thus, we would like to resubmit the updated article for further consideration.

Sincerely,

Kyoungtae Kim

Missouri State University

901 S National Ave.

Springfield, MO, USA 65802

Kkim@missouristate.edu

Reviewer 3 Report

The topic was good. However, there are extensive works suggested to be done.

  1. The reference list was way too short for a review paper. Suggest the authors expand more and recent references. Here are some sentences needed references and the authors can decide more.

Line 79, “RNA-seq has been…”

Line 82, “Furthermore, …”

Sentence on line 84-88.

Line 92, “these modern approaches”

Line 97-100

Line 102-106

Line 114-117

Line 119-121

Line 136 “… are debated and…” If it is debated, can you please provide some references?

Line 139

Line 143

Line 160-161

Line 257-258

Line 260-262

Line 338 “Those studies” which studies?

Line 369-370 and 370-373

Line 524-526 There are a lot of applications for AgNP, I suggest the authors list 2-3 reference for each application. AgNP has been used in antimicrobial membranes as well, suggest the authors cite papers such as: https://www.sciencedirect.com/science/article/abs/pii/S0144861720311905

https://journals.sagepub.com/doi/abs/10.1177/15280837211012590

https://www.mdpi.com/2073-4360/11/12/2057

Line 536-538

Line 644-645

  1. Line 657 has a format issue.
  2. The language is casual for a scientific publication. Please go through the draft and make sure the writing style and wording are professional.

I suggest the authors use consistent tense. Past tense is recommended in this review.

  1. For each session, please reorganize the structure as mechanisms, applications, and case studies. That would be easier to follow, and the review will be a better told story.
  2. The review will be significant if the authors can write a section, and make a table or a diagram to compare the carbon nanotubes, quantum dots and silver nanoparticles.

Author Response

Revisions can be seen on the updated manuscript with track changes. Detailed responses are seen below in blue italicized font. We hope each suggested change is satisfied by our edits and thank you again for the efforts given to improve our manuscript.

Reviewer 3:

The topic was good. However, there are extensive works suggested to be done.

Thank you for your efforts in reviewing our manuscript. We hope each of your comments has been satisfied and feel the manuscript has been greatly improved by your inputs.

The reference list was way too short for a review paper. Suggest the authors expand more and recent references. Here are some sentences needed references and the authors can decide more:

  • Line 79, “RNA-seq has been…”
  • Line 82, “Furthermore, …”
  • Sentence on line 84-88.
  • Line 92, “these modern approaches”
  • Line 97-100
  • Line 102-106
  • Line 114-117
  • Line 119-121
  • Line 136 “… are debated and…” If it is debated, can you please provide some references?
  • Line 139
  • Line 143
  • Line 160-161
  • Line 257-258
  • Line 260-262
  • Line 338 “Those studies” which studies?
  • Line 369-370 and 370-373
  • Line 524-526 There are a lot of applications for AgNP, I suggest the authors list 2-3 reference for each application. AgNP has been used in antimicrobial membranes as well, suggest the authors cite papers such as: https://www.sciencedirect.com/science/article/abs/pii/S0144861720311905
  • https://journals.sagepub.com/doi/abs/10.1177/15280837211012590
  • https://www.mdpi.com/2073-4360/11/12/2057
  • Line 536-538
  • Line 644-645

For each of the lines provided, a needed citation has been added. In some cases, the sentence was deleted to improve the writing. We believe the corrections made improve the citations throughout the manuscript and have greatly lengthened the reference list. Thank you.

Line 657 has a format issue.

The formatting issue at this line has been corrected. Thank you.

The language is casual for a scientific publication. Please go through the draft and make sure the writing style and wording are professional. I suggest the authors use consistent tense. Past tense is recommended in this review.

Throughout the manuscript, the authors have proof read for casual or present tense language. We believe the writing has been greatly improved from this comment. Thank you.

For each session, please reorganize the structure as mechanisms, applications, and case studies. That would be easier to follow, and the review will be a better told story.

We appreciate your comment and input. While the sections have not been reordered in this exact way, the authors have made great adjustments to improve the flow of the information. Writing was added to the introduction section to make for a better story as well. Thank you.

The review will be significant if the authors can write a section, and make a table or a diagram to compare the carbon nanotubes, quantum dots and silver nanoparticles. 

In accordance to this request, the original Table 1 has been replaced with 3 smaller tables that can easily compare the three nanomaterials. We believe this change has made the Tables more efficient and significant. Thank you.

Again, we extend our gratitude for the opportunity to improve our manuscript. We hope our efforts reflect the valuable comments from each reviewer. Thus, we would like to resubmit the updated article for further consideration.

Sincerely,

Kyoungtae Kim

Missouri State University

901 S National Ave.

Springfield, MO, USA 65802

Kkim@missouristate.edu

Reviewer 4 Report

It would be better to specify which work is author's own research and which are other people's work. 

Author Response

Revisions can be seen on the updated manuscript with track changes. Detailed responses are seen below in blue italicized font. We hope each suggested change is satisfied by our edits and thank you again for the efforts given to improve our manuscript.

Reviewer 4:

It would be better to specify which work is author's own research and which are other people's work.

We thank you for your time and effort in editing our work. Corrections have been made to follow your suggestions. It is now specified where the author’s own research is. We believe the manuscript to be improved after following your comments. Thank you.

Again, we extend our gratitude for the opportunity to improve our manuscript. We hope our efforts reflect the valuable comments from each reviewer. Thus, we would like to resubmit the updated article for further consideration.

Sincerely,

Kyoungtae Kim

Missouri State University

901 S National Ave.

Springfield, MO, USA 65802

Kkim@missouristate.edu

Round 2

Reviewer 3 Report

Thank you for the quick edits. I think this review is extraordinary.